# The association of depression with subsequent dementia diagnosis: A Swedish nationwide cohort study from 1964 to 2016

**Sofie Holmquist**[1,2], **Anna Nordström**[1,3], **Peter Nordström**[4]*

**1** Department of Public Health and Clinical Medicine, Occupational and Environmental Medicine, Umeå University, Umeå, Sweden, **2** Department of Applied Educational Science, Umeå University, Umeå, Sweden, **3** Department of Community Medicine, Arctic University of Norway, Tromsø, Norway, **4** Department of Community Medicine and Rehabilitation, Geriatric Medicine, Umeå University, Umeå, Sweden

* peter.nordstrom@.umu.se

**Data Availability Statement:** Data from the Swedish National Patient Register (SNPR) can be requested from the National Board of Health and Welfare in Sweden (www.socialstyrelsen.se). The

## Abstract

### Background

Depression is associated with an increased risk of dementia. However, short follow-up times and lack of adjustment for familial factors in previous studies could influence this association. The purpose of the present study was to investigate the association between depression and subsequent dementia, while controlling for familial factors and with a follow-up of over 35 years.

### Methods and findings

Two cohorts were formed from all individuals aged 50 years or older living in Sweden as of 31 December 2005 ($n = 3,341,010$). The Swedish National Patient Register was searched from 1964 through 2016 to identify diagnosis of depression and dementia. In the first cohort, individuals diagnosed with depression ($n = 119,386$) were matched 1:1 with controls without depression diagnosis. The second cohort was a sibling cohort ($n = 50,644$) consisting of same-sex full sibling pairs with discordant depression status. In the population matched cohort study, a total of 9,802 individuals were diagnosed with dementia during a mean follow-up time of 10.41 (range 0–35) years (5.5% of those diagnosed with depression and 2.6% of those without depression diagnosis (adjusted odds ratio [aOR] 2.47, 95% CI 2.35–2.58; $p < 0.001$), with a stronger association for vascular dementia (aOR 2.68, 95% CI 2.44–2.95; $p < 0.001$) than for Alzheimer disease (aOR 1.79, 95% CI 1.68–1.92; $p < 0.001$). The association with dementia diagnosis was strongest in the first 6 months after depression diagnosis (aOR 15.20, 95% CI 11.85–19.50; $p < 0.001$), then decreased rapidly but persisted over follow-up of more than 20 years (aOR 1.58, 95% CI 1.27–1.98; $p < 0.001$). Also in the sibling cohort, the association was strongest in the first 6 months (aOR 20.85, 95% CI 9.63–45.12; $p < 0.001$), then decreased rapidly but persisted over follow-up of more than 20 years (aOR 2.33, 95% CI 1.32–4.11; $p < 0.001$). The adjusted models included sex, age at baseline, citizenship, civil status, household income, and diagnoses at baseline. The main

background data used in the study can be requested from Statistics Sweden (www.scb.se). To access the data there is a fee to be paid.

**Funding:** The authors did not receive any specific funding for this work.

**Competing interests:** The authors have declared that no competing interests exist.

**Abbreviations:** aOR, adjusted odds ratio; SNPR, Swedish National Patient Register.

limitation of the study methodology is the use of observational data; hence, the associations found are not proof of causal effects.

## Conclusions

Depression is associated with increased odds of dementia, even more than 20 years after diagnosis of depression, and the association remains after adjustment for familial factors. Further research is needed to investigate whether successful prevention and treatment of depression decrease the risk of dementia.

## Author summary

### Why was this study done?

- Dementia is a leading cause of increased need for assistance worldwide among older individuals. The risk of dementia has been associated with previous depression. However, the results from previous studies are not conclusive, there is a lack of studies with long follow-up, and no study has evaluated whether familial factors may influence the association.

### What did the researchers do and find?

- From all inhabitants living in Sweden aged 50 years or older, 2 cohorts were formed: a cohort of 119,386 individuals with depression matched 1:1 with controls without depression, and a cohort of 50,644 full sibling pairs discordant for depression. Both cohorts were evaluated for dementia during follow-up.

- In both cohorts, the risk of dementia was increased 10–20 times in the first year after a diagnosis of depression. Thereafter the risk decreased rapidly but was still evident more than 20 years after the diagnosis of depression.

- The risk of dementia was higher for those with a severe depression compared to those with mild depression, and a stronger association was seen with vascular dementia.

### What do these findings mean?

- The risk of dementia is increased for decades after a diagnosis of depression, where those diagnosed with especially severe depressions are at increased risk.

## Introduction

Dementia is common among the elderly, causing severe individual suffering as well as societal strain [1]. As the proportion of people aged 65 years and above is rapidly increasing in the

world population, the number of individuals with dementia is expected to double within 20 years, and this condition was estimated to have a worldwide cost of US$604 billion in 2010 [2]. Effective treatments for dementia remain scarce [3]; however, a preventive approach may be possible through the identification of high-risk individuals and potentially modifiable risk factors [4–6].

An association between depression and subsequent dementia diagnosis has been suggested in at least 4 independent meta-analyses—indicating close to a 2-fold higher risk of developing dementia following depression [7–10]—as well as several reviews [11–14]. However, there is a lack of studies with large samples and long follow-up, as well as adjustment for important covariates, particularly familial factors. Depression preceding dementia has been discussed both as a risk factor and possible prodrome [9,11], especially with regard to early versus late onset of depression. Thus, several reviews suggest that depression in early life may be a risk factor of subsequent dementia diagnosis [11,13,14] and that depression in late life is rather a prodrome of dementia [11,13], but the literature is not consistent [12]. Some studies have found an association with dementia more than a decade after onset of depression [15], whereas other studies have failed to demonstrate such an association [16,17]. Furthermore, there is evidence suggesting that the association between depression and dementia has a dose–response relationship [12,13,18–20], and the association may be dependent on the type of dementia [13,21,22]. Given the sex differences in the prevalence of depression [23–25] and dementia [26,27], the association may also differ between men and women. Finally, although it is well known that both depression and dementia are influenced by both genetic and familial factors [28–31], it has not been evaluated whether genetic and familial factors influence the association between depression and dementia.

The aim of the present study was to investigate the association between depression and dementia in a large nationwide cohort that included about 120,000 individuals diagnosed with depression and matched controls. Based on a previous study, where we investigated the association between traumatic brain injury and dementia [32], we hypothesized in particular that the association between depression and dementia would persist decades after the depression diagnosis, that the association would be independent of familial factors, and that the association would be stronger with more severe depression for the outcome of vascular dementia.

## Methods

This study is reported as per the Strengthening the Reporting of Observational Studies in Epidemiology (STROBE) guideline (S1 STROBE Checklist).

### Sample

The total cohort from which individuals were considered for inclusion consisted of all Swedish residents ≥50 years old who were alive on 31 December 2005 ($n$ = 3,341,010). A total of 813 individuals were excluded because of missing data on sex, date of birth, or citizenship, or because the date of diagnosis for depression or dementia was postmortem in the national registers used in the present study. From the remaining total cohort ($n$ = 3,340,197), 2 component cohorts were formed and included in the study.

The first cohort consisted of individuals diagnosed with major depression ($n$ = 119,386), with no prior dementia diagnosis, each matched with 1 individual without a diagnosis of depression during follow-up, on the basis of birth year and month, sex, and citizenship (Swedish or non-Swedish). Baseline for both the individuals diagnosed with major depression and their corresponding controls was at the date of depression diagnosis of the individual diagnosed with depression. Controls who died before baseline or were diagnosed with dementia

before baseline were excluded, and a new control was searched from the remaining cohort of potential controls. This procedure was repeated 3 times.

The second cohort was a sibling cohort (*n* = 50,644), consisting of same-sex full sibling pairs with discordant depression status during follow-up. Baseline for both the affected sibling and the unaffected sibling was set at the date of depression diagnosis of the affected sibling. Sibling pairs where the unaffected sibling died before baseline or at least 1 of the siblings was diagnosed with dementia before baseline were excluded. A few sibling pairs had discordant citizenship status. The purpose of the sibling cohort was to control for the effect of early environment and genetic factors (i.e., familial factors).

## Procedure

The Swedish National Patient Register (SNPR), controlled by the National Board of Health and Welfare, was searched from 1 January 1964 through 31 December 2016 to identify diagnosis of major depression and dementia, classified according to the International Classification of Diseases (ICD, 8th, 9th, and 10th revisions). Diagnosis of major depression included the diagnosis codes F32 and F33 (ICD-10), 311 (ICD-9-SWE and ICD-8-SWE), and 296B (ICD-9-SWE). For the F32 and F33 diagnosis codes, depression was coded as mild (F320, F330), moderate (F321, F331), or severe (F322, F323, F332, F333). Depression was considered recurrent for diagnosis code F33 (recurrent depression; ICD-10) or at least 2 diagnoses of depression at least 6 months apart; otherwise, depression was considered as single episode. Diagnosis of dementia was coded as Alzheimer dementia (ICD-10: F00, G30) or vascular dementia (ICD-10: F01) and also included the diagnosis codes F039 (ICD-10) and 290 (ICD-8 and ICD-9). Based on their association with the main exposure in the present study (major depression), the main outcome (dementia), or death, the following covariates were selected and searched in the SNPR using appropriate ICD codes: diabetes, myocardial infarction, alcohol intoxication, drug intoxication, chronic pulmonary disease, renal failure, stroke, and hypothyroidism. The SNPR has national coverage of close to 90% from 1970 to 1986 for inpatient care, and complete coverage since 1987, and in a validation study showed predictive values of 85%–95% for most diagnoses [33]. All Swedish physicians are obliged to report data for all care (except for primary care visits) to the SNPR, including psychiatric care from 1973 and hospital-based outpatient care from 2001 [33]. Diagnosis of death was collected from the Swedish Cause of Death Register, complete since 1961, also controlled by the National Board of Health and Welfare and linked to each individual in the cohort. Data on citizenship (Swedish or non-Swedish), civil status, and household income in 2005 were derived from Statistics Sweden (https://www.scb.se), and linked to each individual in the cohort. There was no written prospective protocol for the analysis; however, all analyses were preplanned based on the results of a previous study [32].

## Statistical analysis

In order to evaluate if the association between diagnosis of depression and subsequent diagnosis of dementia was time dependent, Schoenfeld's residuals were calculated in Stata (version 12.1; StataCorp). Since the proportional hazards assumption was violated, the associations were analyzed in time intervals between baseline and the end of follow-up. For this purpose, multivariable conditional logistic regression analysis was performed in the 2 cohorts, with dementia as the dependent variable. SPSS (version 25.0; IBM) was used for the data analysis. The first model was unadjusted, but by design adjusted for sex, age at baseline, and citizenship in the population matched cohort, and for sex and familial factors in the sibling cohort. A second model was further adjusted for civil status, household income, and diagnoses at baseline

in the population matched cohort, and for citizenship, age at baseline, civil status, household income, and diagnoses at baseline in the sibling cohort (see Table 1). In the population matched cohort study, separate analyses were performed by type of dementia (Alzheimer or vascular dementia) and for different severity (mild, moderate, or severe) and frequency (1 diagnosis of depression or recurrent depression) of depressive episodes. Separate analyses were also performed for men and women. The time dependency of the association between depression and dementia in the 2 different cohorts was graphically illustrated using flexible parametric models with 5 degrees of freedom (Fig 1) [34]. The knots were set in default positions. The differences between individuals based on depression diagnosis at baseline, in the population matched cohort and sibling cohort, were investigated using chi-squared tests for categorical data and independent samples *t* tests for continuous data. For all statistical analyses, significance level was set at $p < 0.05$.

## Ethical considerations

The Regional Ethical Review Board in Umeå (Nr 2013-86-31) and the National Board of Health and Welfare approved the present study. The study was funded by the Swedish Research Council (grant number 2016–02589). The funders had no role in the design, the interpretation of the results, the decision to publish, or the preparation of the manuscript. As data were analyzed anonymously, informed consent was not obtained from study participants.

## Results

### Characteristics of the sample

The characteristics of the population matched cohort and the sibling cohort are displayed in Table 1. Chi-squared test and *t* tests revealed statistically significant differences between cases and controls for civil status, household income, and diagnoses at baseline in the population matched cohort, and for age at baseline, civil status, household income, and diagnoses at baseline in the sibling cohort (all $p < 0.001$; see Table 1).

### Time dependency of the association between depression and subsequent dementia

Schoenfeld's residuals showed that the proportional hazards assumption was violated in the population matched cohort, indicating a time-dependent association between depression and subsequent dementia.

### Incident cases of dementia in the population matched cohort

In the population matched cohort study, a total of 9,802 individuals were diagnosed with dementia during a mean follow-up time of 10.41 (SD 6.89; range 0–35) years (5.7% of those diagnosed with depression and 2.6% of those without depression diagnosis; adjusted odds ratio [aOR] 2.47, 95% CI 2.35–2.58; $p < 0.001$; Table 2). The cumulative incidence of dementia during follow-up is illustrated in Fig 2. The association with dementia diagnosis was strongest in the first 6 months after depression diagnosis (aOR 15.20, 95% CI 11.85–19.50; $p < 0.001$), then decreased rapidly but persisted for more than 20 years (aOR 1.58, 95% CI 1.27–1.98; $p < 0.001$; Table 2).

### Incident cases of dementia in the sibling cohort

In the sibling cohort, a total of 1,161 individuals were diagnosed with dementia during a mean follow-up of 11.04 (SD 6.96; range 0–31) years, 3.4% of those diagnosed with depression and

**Table 1. Cohort characteristics.**

| Characteristic | Matched cohort (n = 238,772) | | | Sibling cohort (n = 50,644) | | |
|---|---|---|---|---|---|---|
| | Depression (n = 119,386) | No depression (n = 119,386) | Group difference: depression versus no depression | Depression (n = 25,322) | No depression (n = 25,322) | Group difference: depression versus no depression |
| **Age at baseline in years, mean (SD)** | 63.79 (11.89) | | | 59.10 (8.85) | 59.97 (8.91) | $t$ (50,639.38) = 11.01, $p$ < 0.001 (2-tailed); 95% CI of the difference = 0.71–1.02 |
| **Citizenship** | | | | | | $\chi^2$ (df = 1, $n$ = 50,644) = 0.026, $p$ = 0.87 (2-sided) |
| Non-Swedish | 42,928 (18.0%) | | | 1,698 (6.7%) | 1,689 (6.7%) | |
| Swedish | 195,844 (82.0%) | | | 23,624 (93.3%) | 23,633 (93.3%) | |
| **Sex** | | | | | | |
| Male | 93,758 (39.3%) | | | 21,416 (42.3%) | | |
| Female | 145,014 (60.7%) | | | 29,228 (57.7%) | | |
| **Civil status** | | | $\chi^2$ (df = 4, $n$ = 238,040) = 4,120.52, $p$ < 0.001 (2-sided) | | | $\chi^2$ (df = 4, $n$ = 55,540) = 815.07, $p$ < 0.001 (2-sided) |
| Married | 53,478 (44.8%) | 67,873 (56.9%) | | 12,108 (47.8%) | 15,165 (59.9%) | |
| Unmarried | 18,009 (15.1%) | 13,934 (11.7%) | | 4,614 (18.2%) | 3,695 (14.6%) | |
| Divorced | 30,333 (25.4%) | 20,600 (17.3%) | | 7,117 (28.1%) | 5,006 (19.8%) | |
| Widowed | 17,297 (14.5%) | 16,394 (13.7%) | | 1,413 (5.7%) | 1,370 (5.4%) | |
| Other | 78 (0.1%) | 44 (<0.1%) | | 21 (0.1%) | 13 (0.1%) | |
| Missing | 191 (0.2%) | 541 (0.5%) | | 31 (0.1%) | 73 (0.3%) | |
| **Yearly household income in Swedish krona, mean (SD); approximate equivalent in US dollars**[*] | 255,808 (339,534); $26,368 (35,350) | 311,620 (489,856); $32,429 (50,977) | $t$ (211,706.04) = 32.31, $p$ < 0.001 (2-tailed); 95% CI of the difference = 52,425.27–59,197.42 | 291,265 (438,927); $30,324 (45,705) | 340,819 (346,022); $5,485 (36,027) | $t$ (47,977.85) = 14.01, $p$ < 0.001 (2-tailed); 95% CI of the difference = 42,664.91–56,443.49 |
| **Diagnoses at baseline** | | | | | | |
| Diabetes | 6,281 (5.3%) | 4,368 (3.7%) | $\chi^2$ (df = 1, $n$ = 238,772) = 359.70, $p$ < 0.001 (2-sided) | 1,172 (4.6%) | 816 (3.2%) | $\chi^2$ (df = 1, $n$ = 50,644) = 66.36, $p$ < 0.001 (2-sided) |
| Myocardial infarction | 5,634 (4.7%) | 4,177 (3.5%) | $\chi^2$ (df = 1, $n$ = 238,772) = 225.65, $p$ < 0.001 (2-sided) | 800 (3.2%) | 644 (2.5%) | $\chi^2$ (df = 1, $n$ = 50,644) = 17.35, $p$ < 0.001 (2-sided) |
| Alcohol intoxication | 7,803 (6.5%) | 1,736 (1.5%) | $\chi^2$ (df = 1, $n$ = 238,772) = 4,019.31, $p$ < 0.001 (2-sided) | 1,977 (7.8%) | 605 (2.4%) | $\chi^2$ (df = 1, $n$ = 50,644) = 768.21, $p$ < 0.001 (2-sided) |
| Drug intoxication | 1,763 (1.5%) | 346 (0.3%) | $\chi^2$ (df = 1, $n$ = 238,772) = 960.54, $p$ < 0.001 (2-sided) | 418 (1.7%) | 115 (0.5%) | $\chi^2$ (df = 1, $n$ = 50,644) = 174.08, $p$ < 0.001 (2-sided) |
| Chronic pulmonary disease | 3,028 (2.5%) | 1,477 (1.2%) | $\chi^2$ (df = 1, $n$ = 238,772) = 544.25, $p$ < 0.001 (2-sided) | 594 (2.3%) | 315 (1.2%) | $\chi^2$ (df = 1, $n$ = 50,644) = 87.20, $p$ < 0.001 (2-sided) |
| Renal failure | 1,325 (1.1%) | 708 (0.6%) | $\chi^2$ (df = 1, $n$ = 238,772) = 188.86, $p$ < 0.001 (2-sided) | 245 (1.0%) | 122 (0.5%) | $\chi^2$ (df = 1, $n$ = 50,644) = 41.52., $p$ < 0.001 (2-sided) |
| Stroke | 5,461 (4.6%) | 2,967 (2.5%) | $\chi^2$ (df = 1, $n$ = 238,772) = 765.02, $p$ < 0.001 (2-sided) | 857 (3.4%) | 454 (1.8%) | $\chi^2$ (df = 1, $n$ = 50,644) = 127.17, $p$ < 0.001 (2-sided) |
| Hypothyroidism | 2,622 (2.2%) | 1,589 (1.3%) | $\chi^2$ (df = 1, $n$ = 238,772) = 257.95, $p$ < 0.001 (2-sided) | 495 (2.0%) | 299 (1.2%) | $\chi^2$ (df = 1, $n$ = 50,644) = 49.15, $p$ < 0.001 (2-sided) |
| **Survival time in years, mean (SD)** | 10.04 (6.91) | 10.79 (6.86) | $t$ (238,754.64) = 26.80, $p$ < 0.001 (2-tailed); 95% CI of the difference = 0.70–0.81 | 10.78 (7.00) | 11.30 (6.90) | $t$ (50,631.57) = 8.39, $p$ < 0.001 (2-tailed); 95% CI of the difference = 0.40–0.64 |

Data are number (percent) unless otherwise indicated. Information on citizenship, civil status, and household income was derived from Statistics Sweden for the year of 2005.

[*]Using 2019 exchange rate: 1:0.104.

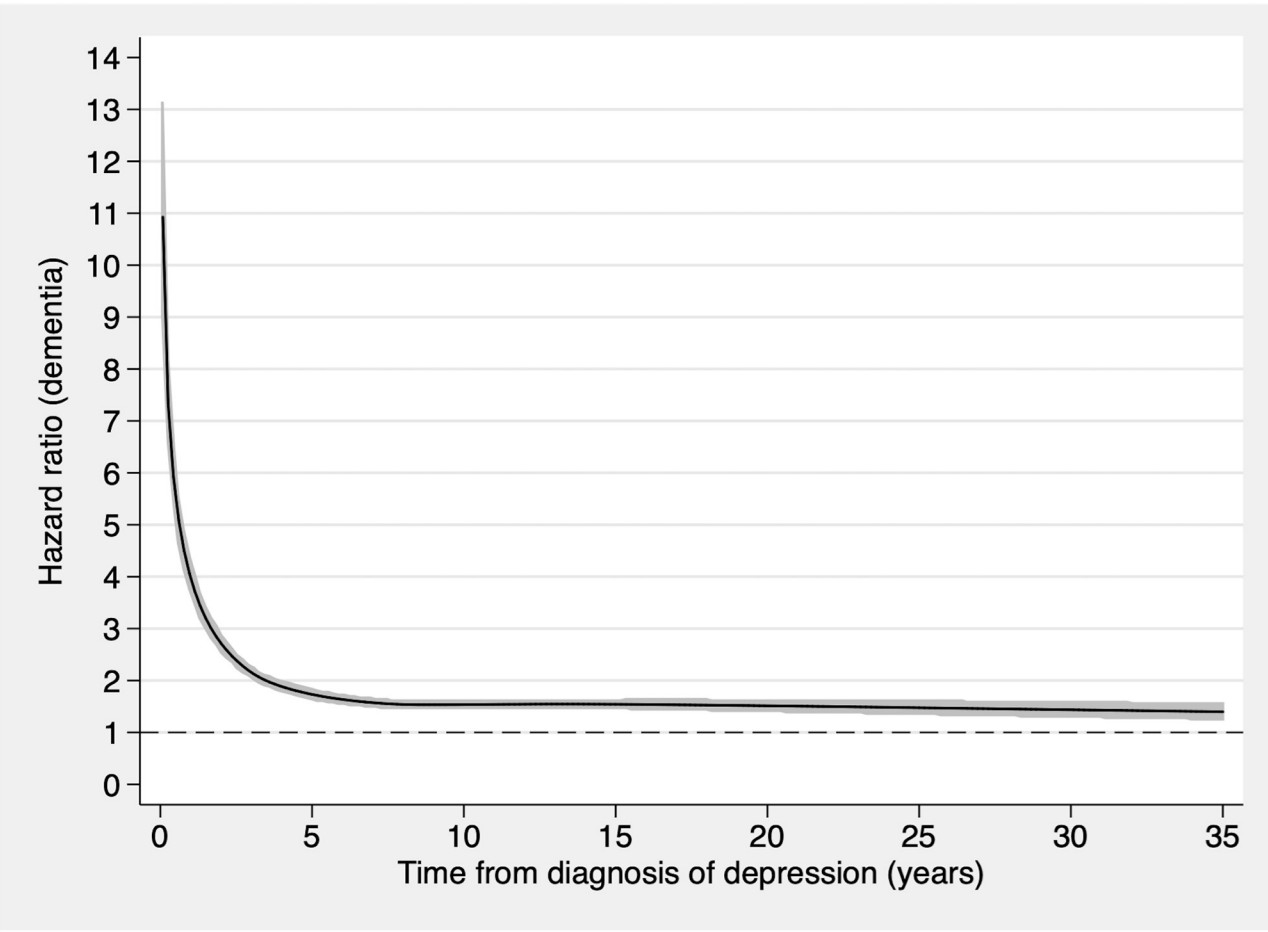

**Fig 1. Association between depression and the risk of dementia during follow-up for 238,772 individuals.** Individuals with follow-up time of less than 1 month were excluded. The figure was constructed using a flexible parametric model with 5 degrees of freedom and knots in default positions. The black line represents the hazard ratio, and the grey areas represent 95% confidence intervals.

1.2% of those without depression diagnosis (aOR 3.75, 95% CI 3.24–4.34; $p$ < 0.001; Table 3). The cumulative incidence of dementia during follow-up in the sibling cohort is illustrated in Fig 3. As in the population matched cohort, the association with dementia diagnosis was strongest the first 6 months after depression diagnosis (aOR 20.85, 95% CI 9.63–45.12; $p$ < 0.001), then decreased rapidly but persisted over follow-up of more than 20 years (aOR 2.33, 95% CI 1.32–4.11; $p$ < 0.001; Table 3).

## Severity and frequency of depression, type of dementia, and patient sex in the population matched cohort

The severity of the depression influenced the association with dementia (Table 4). Thus, after a mild depression, the risk of subsequent dementia remained elevated 5–9.9 years after depression diagnosis (aOR 1.96, 95% CI 1.39–2.75; $p$ < 0.001) but not at ≥10 years after depression diagnosis (aOR 1.39, 95% CI 0.83–2.36; $p$ = 0.212). In contrast, the association remained significant at ≥10 years after depression diagnosis for both for moderate depression (aOR 1.41, 95% CI 1.02–1.95; $p$ = 0.037) and for severe depression (aOR 1.75, 95% CI 1.29–2.36; $p$ < 0.001). No clear differences were found between a single episode of depression and recurrent

**Table 2. Associations between depression and the risk of subsequent dementia diagnosis during follow-up in 119,386 individuals diagnosed with depression and 119,386 matched controls without depression diagnosis: Population matched cohort (n = 238,772).**

| Time after depression diagnosis | Individuals at risk | Diagnosed with dementia | | | | | Unadjusted* | | | Adjusted** | | |
|---|---|---|---|---|---|---|---|---|---|---|---|---|
| | | Total | Cases | | Controls | | OR | 95% CI | p-Value | OR | 95% CI | p-Value |
| | | | Count | Percent | Count | Percent | | | | | | |
| Overall | 238,772 | 9,802 | 6,752 | 5.66% | 3,050 | 2.55% | 2.37 | 2.27–2.48 | <0.001 | 2.47 | 2.35–2.58 | <0.001 |
| 0–5.9 months | 238,772 | 1,043 | 972 | 0.81% | 71 | 0.06% | 14.06 | 11.01–17.95 | <0.001 | 15.20 | 11.85–19.50 | <0.001 |
| 6–11.9 months | 232,274 | 750 | 652 | 0.57% | 98 | 0.08% | 7.24 | 5.80–9.03 | <0.001 | 7.83 | 6.24–9.83 | <0.001 |
| 1–1.9 years | 226,605 | 1,006 | 816 | 0.73% | 190 | 0.17% | 4.48 | 4.09–5.73 | <0.001 | 5.15 | 4.33–6.12 | <0.001 |
| 2–4.9 years | 215,680 | 2,129 | 1,474 | 1.40% | 655 | 0.59% | 2.58 | 2.34–2.86 | <0.001 | 2.62 | 2.36–2.91 | <0.001 |
| 5–9.9 years | 180,192 | 2,289 | 1,384 | 1.59% | 914 | 0.98% | 1.83 | 1.67–2.01 | <0.001 | 1.85 | 1.68–2.05 | <0.001 |
| 10–20 years | 112,027 | 2,095 | 1,187 | 2.21% | 908 | 1.55% | 1.54 | 1.40–1.70 | <0.001 | 1.58 | 1.43–1.75 | <0.001 |
| 20 years or more | 26,044 | 481 | 267 | 2.19% | 214 | 1.55% | 1.60 | 1.30–1.97 | <0.001 | 1.58 | 1.27–1.98 | <0.001 |

ORs and 95% confidence intervals were derived from conditional logistic regression analysis.

*By design adjusted for sex, age at baseline, and citizenship.

**Further adjusted for civil status, household income, and diagnoses at baseline: diabetes, myocardial infarction, alcohol intoxication, drug intoxication, chronic pulmonary disease, renal failure, stroke, and hypothyroidism.

OR, odds ratio.

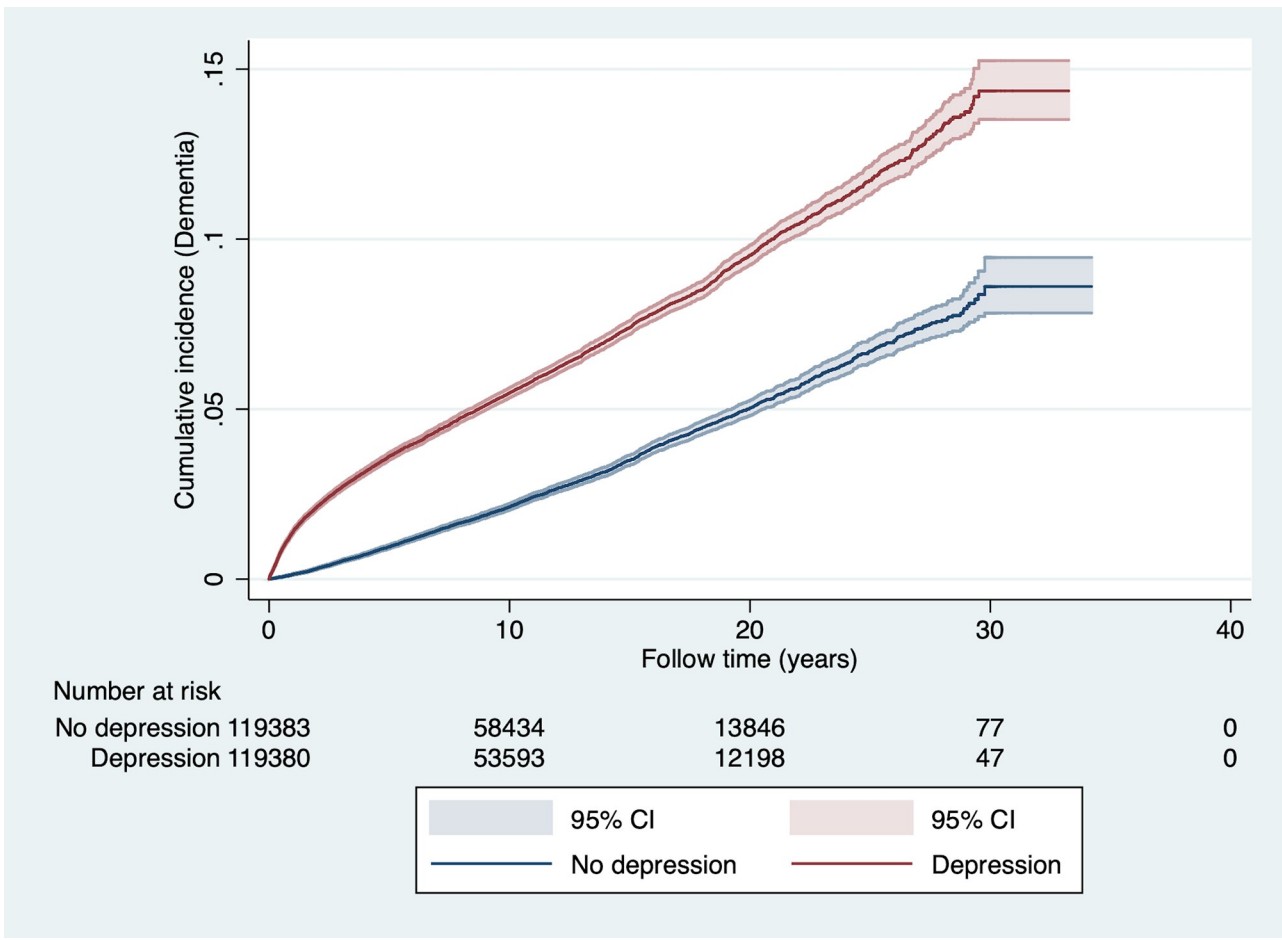

**Fig 2. Cumulative incidence of dementia after depression diagnosis in the matched cohort of 119,386 individuals with depression and 119,386 individuals with no depression during follow-up.** The number of individuals at risk at each time point is presented below the graph.

**Table 3. Associations between depression and the risk of subsequent dementia diagnosis during follow-up for 25,322 full sibling pairs with discordant depression status at baseline: Sibling cohort (*n* = 50,644).**

| Time after depression diagnosis | Individuals at risk | Diagnosed with dementia | | | | | Unadjusted* | | | Adjusted** | | |
|---|---|---|---|---|---|---|---|---|---|---|---|---|
| | | Total | Cases | | Controls | | OR | 95% CI | *p*-Value | OR | 95% CI | *p*-Value |
| | | | Count | Percent | Count | Percent | | | | | | |
| Overall | 50,644 | 1,161 | 864 | 3.41% | 297 | 1.17% | 3.16 | 2.75–3.64 | <0.001 | 3.75 | 3.24–4.34 | <0.001 |
| 0–5.9 months | 50,644 | 135 | 128 | 0.51% | 7 | 0.03% | 18.29 | 8.55–39.13 | <0.001 | 20.85 | 9.63–45.12 | <0.001 |
| 6–11.9 months | 49,600 | 86 | 82 | 0.33% | 4 | 0.02% | 20.25 | 7.42–55.26 | <0.001 | 24.39 | 8.82–67.45 | <0.001 |
| 1–1.9 years | 48,609 | 136 | 116 | 0.48% | 20 | 0.08% | 5.80 | 3.61–9.32 | <0.001 | 7.07 | 4.33–11.54 | <0.001 |
| 2–4.9 years | 46,666 | 228 | 174 | 0.76% | 54 | 0.23% | 3.49 | 2.54–4.79 | <0.001 | 4.25 | 3.04–5.94 | <0.001 |
| 5–9.9 years | 39,894 | 248 | 156 | 0.80% | 92 | 0.45% | 1.86 | 1.41–2.45 | <0.001 | 2.03 | 1.51–2.72 | <0.001 |
| 10–20 years | 25,708 | 252 | 159 | 1.27% | 93 | 0.70% | 1.96 | 1.49–2.58 | <0.001 | 2.50 | 1.86–3.35 | <0.001 |
| 20 years or more | 6,296 | 76 | 49 | 1.61% | 27 | 0.83% | 1.95 | 1.15–3.30 | 0.013 | 2.33 | 1.32–4.11 | <0.001 |

ORs and confidence intervals were derived from conditional logistic regression analysis.

*By use of a sibling design, adjusted for sex and familial factors.

**Further adjusted for citizenship, age at baseline, civil status, household income, and diagnoses at baseline: diabetes, myocardial infarction, alcohol intoxication, drug intoxication, chronic pulmonary disease, renal failure, stroke, and hypothyroidism.

OR, odds ratio.

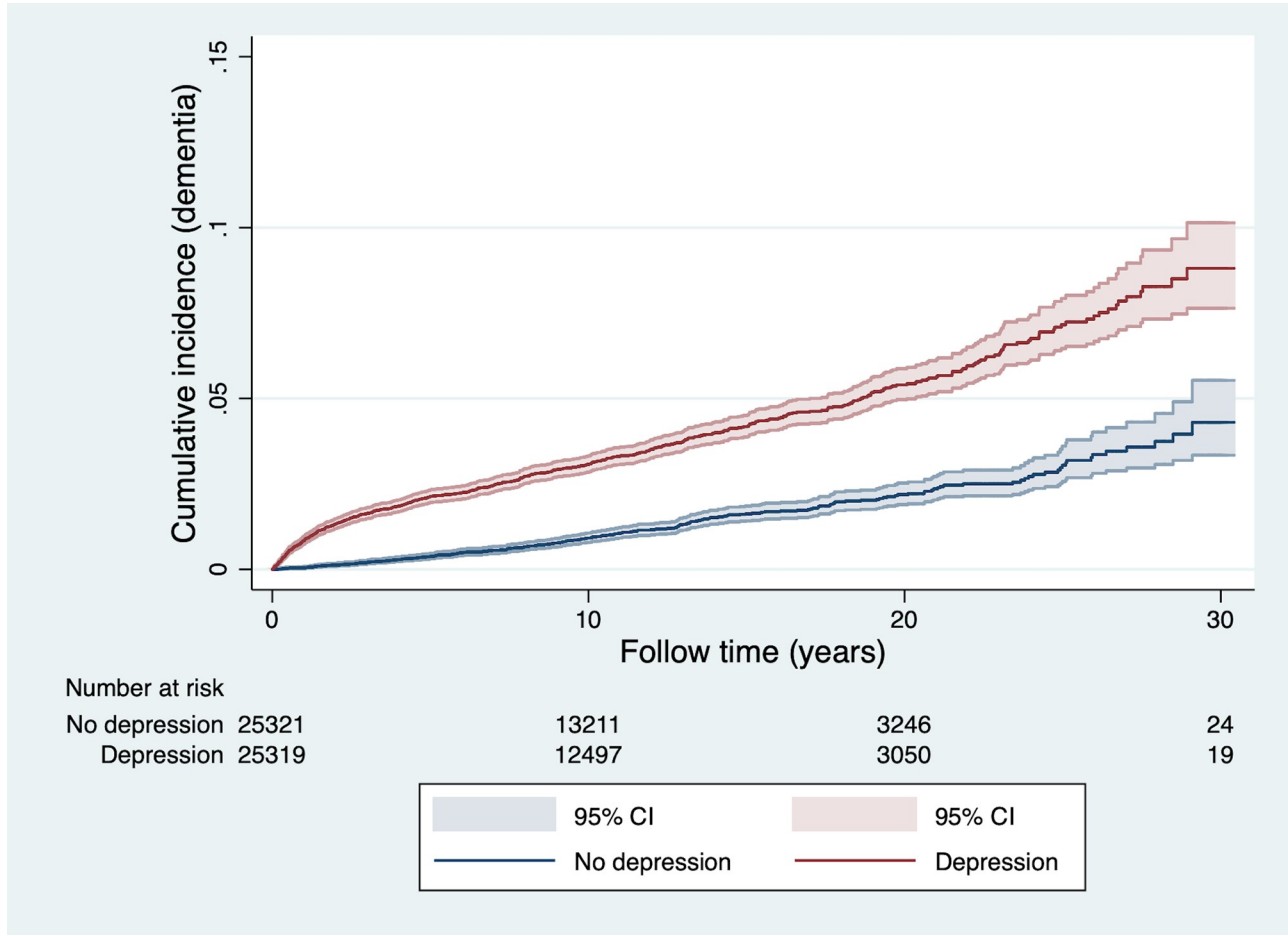

**Fig 3. Cumulative incidence of dementia after depression diagnosis in 25,322 sibling pairs discordant for depression diagnosis.** The number of individuals at risk at each time point is presented below the graph.

**Table 4. The risk of dementia diagnosis during follow-up for individuals diagnosed with mild depression (n = 10,030), moderate depression (n = 24,746), and severe depression (n = 20,540), compared to matched controls.**

| Time after depression diagnosis | Individuals at risk | Diagnosed with dementia | Unadjusted* | | | Adjusted** | | |
|---|---|---|---|---|---|---|---|---|
| | | | OR | 95% CI | p-Value | OR | 95% CI | p-Value |
| **Mild depression (n = 20,060)** | | | | | | | | |
| Overall | 20,060 | 728 | 2.75 | 2.32–3.26 | <0.001 | 2.85 | 2.38–3.39 | <0.001 |
| 0–5.9 months | 20,060 | 75 | 17.75 | 6.48–48.60 | <0.001 | 18.02 | 6.53–49.71 | <0.001 |
| 6–11.9 months | 19,495 | 69 | 5.70 | 2.91–11.16 | <0.001 | 5.61 | 2.83–11.11 | <0.001 |
| 1–1.9 years | 18,921 | 82 | 4.57 | 2.56–8.15 | <0.001 | 4.26 | 2.33–7.80 | <0.001 |
| 2–4.9 years | 17,840 | 222 | 3.74 | 2.66–5.25 | <0.001 | 3.98 | 2.79–5.68 | <0.001 |
| 5–9.9 years | 14,269 | 191 | 1.91 | 1.38–2.65 | <0.001 | 1.96 | 1.39–2.75 | <0.001 |
| 10+ years | 7,117 | 89 | 1.33 | 0.81–2.20 | 0.258 | 1.39 | 0.83–2.36 | 0.212 |
| **Moderate depression (n = 49,492)** | | | | | | | | |
| Overall | 49,492 | 1,426 | 2.07 | 1.84–2.32 | <0.001 | 2.16 | 1.92–2.43 | <0.001 |
| 0–5.9 months | 49,492 | 124 | 11.40 | 5.97–21.76 | <0.001 | 11.66 | 6.06–22.43 | <0.001 |
| 6–11.9 months | 48,309 | 108 | 5.63 | 3.31–9.57 | <0.001 | 5.93 | 3.46–10.18 | <0.001 |
| 1–1.9 years | 47,092 | 178 | 3.50 | 2.44–5.02 | <0.001 | 3.70 | 2.56–5.36 | <0.001 |
| 2–4.9 years | 44,802 | 367 | 2.14 | 1.70–2.69 | <0.001 | 2.16 | 1.71–2.74 | <0.001 |
| 5–9.9 years | 36,401 | 417 | 1.58 | 1.27–1.96 | <0.001 | 1.56 | 1.25–1.95 | <0.001 |
| 10+ years | 19,177 | 229 | 1.36 | 1.00–1.85 | 0.053 | 1.41 | 1.02–1.95 | 0.037 |
| **Severe depression (n = 41,080)** | | | | | | | | |
| Overall | 41,080 | 1,479 | 2.32 | 2.07–2.61 | <0.001 | 2.36 | 2.10–2.66 | <0.001 |
| 0–5.9 months | 41,080 | 127 | 20.17 | 8.88–45.78 | <0.001 | 20.99 | 9.21–47.89 | <0.001 |
| 6–11.9 months | 39,951 | 130 | 6.38 | 3.76–10.80 | <0.001 | 6.77 | 3.96–11.58 | <0.001 |
| 1–1.9 years | 38,896 | 152 | 6.42 | 3.96–10.41 | <0.001 | 6.08 | 3.71–9.97 | <0.001 |
| 2–4.9 years | 36,939 | 396 | 2.32 | 1.85–2.90 | <0.001 | 2.43 | 1.92–3.06 | <0.001 |
| 5–9.9 years | 30,558 | 403 | 1.78 | 1.43–2.22 | <0.001 | 1.77 | 1.41–2.22 | <0.001 |
| 10+ years | 17,095 | 281 | 1.80 | 1.35–2.39 | <0.001 | 1.75 | 1.29–2.36 | <0.001 |

Classification of the severity of the depression based of the International Classification of Diseases–10th revision. ORs and confidence intervals were derived from conditional logistic regression analysis.

*By design adjusted for patient sex, age at baseline, and citizenship.

**Further adjusted for civil status, household income, and diagnoses at baseline: diabetes, myocardial infarction, alcohol intoxication, drug intoxication, chronic pulmonary disease, renal failure, stroke, and hypothyroidism.

OR, odds ratio.

depression or between men and women (Table 5). The type of dementia also influenced the magnitude of the association. The association was stronger for vascular dementia than for Alzheimer disease, and remained at ≥20 years only for vascular dementia (aOR 1.64, 95% CI 1.02–2.62; p = 0.040, Table 5).

## Discussion

The results of the present nationwide study indicated increased odds for dementia diagnosis after depression, even when the depression occurred 20 years or more before the dementia diagnosis and while adjusting for a set of covariates including sex, age at baseline, citizenship, civil status, household income, and diagnoses at baseline. The association was strongest when dementia occurred within the first 6 months after depression diagnosis, and then decreased rapidly but remained statistically significant, with an OR of close to 1.6 at a follow-up time of 20 years or more. The results were consistent when controlling for familial factors in a

**Table 5. Risk of dementia diagnosis during follow-up for individuals with diagnosed single depression (n = 56,522) and recurrent depression (n = 62,864), for men with depression (n = 46,879) and women with depression (n = 75,507), and for Alzheimer disease and vascular dementia separately, compared to matched controls.**

| Time after depression diagnosis | Individuals at risk | Diagnosed with dementia | Unadjusted | | | Adjusted | | |
|---|---|---|---|---|---|---|---|---|
| | | | OR | 95% CI | p-Value | OR | 95% CI | p-Value |
| **Single depressive episode (n = 113,044)\*** | | | | | | | | |
| Overall | 113,044 | 4,992 | 2.32 | 2.18–2.48 | <0.001 | 2.48 | 2.32–2.65 | <0.001 |
| 0–5.9 months | 113,044 | 837 | 16.77 | 12.49–22.50 | <0.001 | 18.82 | 13.92–25.45 | <0.001 |
| 6–11.9 months | 107,693 | 506 | 7.45 | 5.66–9.80 | <0.001 | 8.17 | 6.17–10.82 | <0.001 |
| 1–1.9 years | 103,787 | 547 | 4.42 | 3.53–5.52 | <0.001 | 5.06 | 4.01–6.39 | <0.001 |
| 2–4.9 years | 97,340 | 1,070 | 2.46 | 2.13–2.84 | <0.001 | 2.63 | 2.26–3.06 | <0.001 |
| 5–9.9 years | 78,049 | 987 | 1.51 | 1.31–1.74 | <0.001 | 1.51 | 1.30–1.75 | <0.001 |
| 10–19.9 years | 47,002 | 839 | 1.13 | 0.97–1.32 | 0.108 | 1.14 | 0.97–1.34 | 0.107 |
| 20 years or more | 12,736 | 206 | 1.56 | 1.13–2.15 | 0.007 | 1.61 | 1.14–2.26 | 0.006 |
| **Recurrent depression (n = 125,728)\*** | | | | | | | | |
| Overall | 125,728 | 4,810 | 2.43 | 2.27–2.59 | <0.001 | 2.47 | 2.31–2.64 | <0.001 |
| 0–5.9 months | 125,728 | 206 | 8.27 | 5.32–12.88 | <0.001 | 8.47 | 5.41–13.27 | <0.001 |
| 6–11.9 months | 124,581 | 244 | 6.84 | 4.69–9.97 | <0.001 | 7.54 | 5.11–11.13 | <0.001 |
| 1–1.9 years | 122,818 | 459 | 5.41 | 4.20–6.99 | <0.001 | 5.38 | 4.15–6.99 | <0.001 |
| 2–4.9 years | 118,340 | 1,059 | 2.70 | 2.35–3.11 | <0.001 | 2.66 | 2.31–3.07 | <0.001 |
| 5–9.9 years | 102,143 | 1,311 | 2.10 | 1.86–2.38 | <0.001 | 2.14 | 1.87–2.44 | <0.001 |
| 10–19.9 years | 65,025 | 1,256 | 1.90 | 1.67–2.16 | <0.001 | 1.96 | 1.72–2.23 | <0.001 |
| 20 years or more | 13,308 | 275 | 1.63 | 1.24–2.15 | <0.001 | 1.55 | 1.16–2.09 | <0.001 |
| **Men (n = 93,758)\*\*** | | | | | | | | |
| Overall | 90,375 | 3,383 | 2.52 | 2.33–2.72 | <0.001 | 2.63 | 2.43–2.85 | <0.001 |
| 0–5.9 months | 90,375 | 380 | 17.90 | 11.41–28.08 | <0.001 | 19.59 | 12.32–30.98 | <0.001 |
| 6–11.9 months | 90,855 | 284 | 7.18 | 5.01–10.27 | <0.001 | 7.92 | 5.48–11.44 | <0.001 |
| 1–1.9 years | 89,239 | 378 | 4.68 | 3.57–6.15 | <0.001 | 4.87 | 3.67–6.46 | <0.001 |
| 2–4.9 years | 83,664 | 725 | 2.83 | 2.37–3.37 | <0.001 | 2.81 | 2.34–3.38 | <0.001 |
| 5–9.9 years | 68,972 | 777 | 2.07 | 1.75–2.44 | <0.001 | 2.06 | 1.73–2.45 | <0.001 |
| 10–19.9 years | 42,485 | 693 | 1.68 | 1.42–1.99 | <0.001 | 1.78 | 1.49–2.12 | <0.001 |
| 20 years or more | 9,752 | 146 | 1.46 | 1.01–2.12 | 0.050 | 1.41 | 0.95–2.11 | 0.091 |
| **Women (n = 145,014)\*\*** | | | | | | | | |
| Overall | 145,014 | 6,419 | 2.30 | 2.18–2.43 | <0.001 | 2.38 | 2.25–2.52 | <0.001 |
| 0–5.9 months | 145,014 | 663 | 12.49 | 9.34–16.71 | <0.001 | 13.42 | 9.97–18.05 | <0.001 |
| 6–11.9 months | 144,419 | 446 | 7.27 | 5.49–9.64 | <0.001 | 7.83 | 5.86–10.45 | <0.001 |
| 1–1.9 years | 138,312 | 628 | 4.94 | 3.99–6.12 | <0.001 | 5.35 | 4.30–6.67 | <0.001 |
| 2–4.9 years | 132,016 | 1,404 | 2.47 | 2.19–2.79 | <0.001 | 2.54 | 2.24–2.89 | <0.001 |
| 5–9.9 years | 111,220 | 1,521 | 1.73 | 1.54–1.93 | <0.001 | 1.76 | 1.57–1.98 | <0.001 |
| 10–19.9 years | 69,542 | 1,402 | 1.48 | 1.31–1.66 | <0.001 | 1.50 | 1.33–1.69 | <0.001 |
| 20 years or more | 16,292 | 335 | 1.67 | 1.30–2.15 | <0.001 | 1.66 | 1.27–2.17 | <0.001 |
| **Alzheimer disease (n = 238,772)\*** | | | | | | | | |
| Overall | 238,772 | 4,201 | 1.65 | 1.54–1.75 | <0.001 | 1.79 | 1.68–1.92 | <0.001 |
| 0–5.9 months | 238,772 | 470 | 10.19 | 7.42–13.99 | <0.001 | 11.32 | 8.20–15.61 | <0.001 |
| 6–11.9 months | 232,274 | 343 | 5.66 | 4.19–7.65 | <0.001 | 6.39 | 4.69–8.71 | <0.001 |
| 1–1.9 years | 226,605 | 462 | 3.96 | 3.13–4.99 | <0.001 | 4.56 | 3.58–5.80 | <0.001 |
| 2–4.9 years | 215,680 | 918 | 1.94 | 1.68–2.24 | <0.001 | 2.13 | 1.84–2.47 | <0.001 |
| 5–9.9 years | 180,192 | 978 | 1.21 | 1.05–1.38 | 0.007 | 1.28 | 1.11–1.47 | 0.001 |
| 10–19.9 years | 112,027 | 825 | 0.89 | 0.77–1.04 | 0.13 | 0.97 | 0.83–1.23 | 0.657 |
| 20 years or more | 26,044 | 205 | 0.99 | 0.73–1.34 | 0.938 | 0.97 | 0.70–1.34 | 0.854 |

*(Continued)*

**Table 5.** (Continued)

| Time after depression diagnosis | Individuals at risk | Diagnosed with dementia | Unadjusted | | | Adjusted | | |
|---|---|---|---|---|---|---|---|---|
| | | | OR | 95% CI | *p*-Value | OR | 95% CI | *p*-Value |
| **Vascular dementia (*n* = 238,772)*** | | | | | | | | |
| Overall | 238,772 | 2,329 | 2.74 | 2.50–3.01 | <0.001 | 2.68 | 2.44–2.95 | <0.001 |
| 0–5.9 months | 238,772 | 254 | 20.17 | 11.30–36.01 | <0.001 | 21.19 | 11.51–39.00 | <0.001 |
| 6–11.9 months | 232,274 | 188 | 9.17 | 5.64–14.91 | <0.001 | 9.54 | 5.81–15.66 | <0.001 |
| 1–1.9 years | 226,605 | 241 | 6.00 | 4.15–8.67 | <0.001 | 5.61 | 3.84–8.17 | <0.001 |
| 2–4.9 years | 215,680 | 523 | 3.06 | 2.48–3.77 | <0.001 | 2.72 | 2.19–3.37 | <0.001 |
| 5–9.9 years | 180,192 | 529 | 2.37 | 1.93–2.90 | <0.001 | 2.36 | 1.91–2.91 | <0.001 |
| 10–19.9 years | 112,027 | 491 | 1.72 | 1.42–2.10 | <0.001 | 1.69 | 1.37–2.07 | <0.001 |
| 20 years or more | 26,044 | 103 | 1.58 | 1.02–2.44 | 0.041 | 1.64 | 1.02–2.62 | 0.040 |

Depression was considered as recurrent for diagnosis code F33 (recurrent depression; ICD-10) or at least 2 diagnoses of depression at least 6 months apart; otherwise, depression was considered as a single episode. ORs and confidence intervals were derived from conditional logistic regression analysis.

*Unadjusted model is by design adjusted for patient sex, age at baseline, and citizenship. Adjusted model is further adjusted for civil status, household income, and diagnoses at baseline: diabetes, myocardial infarction, alcohol intoxication, drug intoxication, chronic pulmonary disease, renal failure, stroke, and hypothyroidism.

**Unadjusted model is by design adjusted for age at baseline and citizenship. Adjusted model is further adjusted for civil status, household income, and diagnoses at baseline: diabetes, myocardial infarction, alcohol intoxication, drug intoxication, chronic pulmonary disease, renal failure, stroke, and hypothyroidism.

OR, odds ratio.

nationwide sibling cohort. The results supported our hypothesis of an association between depression and dementia persisting decades after the depression diagnosis date, independent of familial factors.

Although an association between depression and risk of subsequent dementia has been established previously [7,8,11,12,35], the nature of the association has not been explored in great detail. In studies with short follow-up times, the association may reflect several different potential underlying relationships: depressive symptoms might be a characteristic of the prodromal phase of dementia, the patient might in fact already have dementia although yet undiagnosed, or the depression might be discovered and diagnosed when the patient is under evaluation for dementia. Additionally, lack of evaluation of important covariates increases the risk of confounding and bias. In the present study, the maximum follow-up time was 35 years, and the association between depression and subsequent risk of dementia was consistent for more than 20 years of follow-up. In addition, the association remained after adjusting for an array of important covariates including medical conditions and socioeconomic factors. The results were similar for men and women, suggesting an association independent of patient sex.

To the best of our knowledge, it has not been previously established whether the association between dementia and depression remains after controlling for familial factors. This is of importance as early environmental or genetic factors may mediate the association between depression and subsequent dementia. Interestingly, in a previous study using a twin design, there was an association between depression and dementia [16]. However, 40% of the dementia diagnoses occurred within 4 years of the depression, and the study sample was small, which limited the possibility of evaluating the association during longer follow-up times; the study also lacked covariates. Thus, the association found may well represent depression in individuals with undiagnosed dementia, or dementia in an early phase. In the present study we controlled for familial factors using a sibling design including more than 25,000 sibling pairs. The results from the sibling cohort showed an association between depression and subsequent risk of dementia that lasted for more than 20 years. Thus, the association was similar to that found

in the population matched cohort study, suggesting an association between the 2 diagnoses independent of genetic and early environmental factors.

It is also not clear from previous studies whether the association between depression and dementia is dependent on dementia type [13,21,22]. Our results indicate that the association is stronger and covers a wider timespan for vascular dementia than for Alzheimer disease: For vascular dementia, the risk remained statistically significant over a follow-up of more than 20 years, whereas for Alzheimer disease the association remained statistically significant for less than 10 years. The association of depression with Alzheimer disease may best be described as reflecting prodromal depressive symptoms [36]. With respect to the stronger association with vascular dementia, it could be that depression induces the inflammatory process of atherosclerosis that is known to begin already early in life [37]. This hypothesis has support from a recent study in which Setiawan and others [38] found signs of increased activation of microglia in certain areas of the brain in individuals after a major depression, as well as from a study noting cerebrovascular disease in depressed elderly individuals to be associated with cognitive decline [39]. Yet, given the observational design of these studies and our study, no causal inferences can be made. Previous studies [18–20] and reviews [12,13] have suggested a dose–response relationship between depression and the risk of subsequent dementia. In the present study the association between depression and risk of subsequent dementia remained statistically significant for moderate and severe depression for a follow-up of more than 10 years, in contrast to mild depression. However, the association was similar for a single episode of depression and recurrent depression. Thus, our hypothesis of a dose–response relationship between depression and subsequent dementia was only partly supported. Again, whether this dose–response relationship indicates a causal relationship cannot be determined in the present study.

Thus, a limitation of the present study is the use of observational data. Possible explanations for the found association between depression and dementia include depression as a risk factor for dementia, and depression and dementia being associated based on meditating pathogenesis. In particular, there is increased risk of reverse causality for those diagnosed with dementia shortly after a depression diagnosis, where the association may reflect depressive symptoms as a prodrome of yet-undiagnosed dementia, or individuals who are evaluated for dementia also being diagnosed with depression due to more intense healthcare. Moreover, even though we controlled for civil status and household income, we were not able to account for additional socioeconomic factors, e.g., education level. Additionally, since the SNPR doesn't include data from general practice, some of the individuals included in the study may have been diagnosed with either dementia or depression in a primary care setting. Additionally, a limitation of our study is that the results here may not necessarily generalize to individuals who were diagnosed in a primary care setting because the individuals included in the study were diagnosed in specialty care, and as such may represent more severe cases of depression/dementia. This may also explain the lower than expected rates of depression and dementia in our study, since many patients are diagnosed exclusively in primary care: Previous studies have indicated that the prevalence of both depression and dementia is around 10% in older community-dwelling individuals [40,41]. Nevertheless, our results distinctly reveal a long-lasting association between depression and subsequent risk of developing dementia, information that may be useful in the identification of individuals potentially at high risk for development of dementia in late life. A potential limitation in the present study is the use of ORs instead of risk ratios, since the use of ORs can result in non-collapsibility, i.e., that estimated ORs are not similar to the relative risks. However, this potential problem seems to be of minor importance as the estimated relative risks and ORs for the different time intervals were similar in the present study. Strengths of the present study include the use of clinically confirmed diagnoses of depression and dementia made in specialist care. Other strengths include the nationwide cohort studied,

the long follow-up of up to 35 years, and the use of a sibling design to control for familial factors, all together increasing the external validity.

In conclusion, depression diagnosis is associated with increased odds of subsequent dementia diagnosis, even when the depression diagnosis occurs more than 20 years before the dementia diagnosis. The association was stronger for vascular dementia than for Alzheimer disease, which might suggest that depression contributes to the inflammatory process characteristic of atherosclerosis specifically in the brain. It would be of interest if future longitudinal studies could evaluate whether any inflammatory process associated with depression is reduced locally in the brain if depression is successfully treated, and whether the successful treatment of a depressive episode reduces the later risk of dementia.

## Supporting information

**S1 STROBE Checklist.**
(DOCX)

## Author Contributions

**Conceptualization:** Anna Nordström, Peter Nordström.

**Data curation:** Sofie Holmquist, Peter Nordström.

**Formal analysis:** Sofie Holmquist, Peter Nordström.

**Investigation:** Peter Nordström.

**Methodology:** Sofie Holmquist.

**Supervision:** Anna Nordström, Peter Nordström.

**Writing – original draft:** Sofie Holmquist, Peter Nordström.

**Writing – review & editing:** Sofie Holmquist, Anna Nordström, Peter Nordström.

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
