## [Decision Letter · Decision Letter 0]

16 Oct 2019

Dear Dr. Nordström,

Thank you very much for submitting your manuscript "Depression and the risk of subsequent dementia diagnosis: A Nationwide Cohort Study" (PMEDICINE-D-19-03130) for consideration at PLOS Medicine. 

Your paper was evaluated by a senior editor and discussed among all the editors here. It was also discussed with an academic editor with relevant expertise, and sent to three independent reviewers, including a statistical reviewer. The reviews are appended at the bottom of this email and any accompanying reviewer attachments can be seen via the link below:

[LINK]

In light of these reviews, I am afraid that we will not be able to accept the manuscript for publication in the journal in its current form, but we would like to consider a revised version that addresses the reviewers' and editors' comments. Obviously we cannot make any decision about publication until we have seen the revised manuscript and your response, and we plan to seek re-review by one or more of the reviewers. 

We expect to receive your revised manuscript by Nov 06 2019 11:59PM. Please email us (plosmedicine@plos.org) if you have any questions or concerns.

We look forward to receiving your revised manuscript. 

Sincerely,

Caitlin Moyer, Ph.D.

Associate Editor 

PLOS Medicine

plosmedicine.org

1. Did your study have a prospective protocol or analysis plan? Please state this (either way) early in the Methods section.

c) In either case, changes in the analysis—including those made in response to peer review comments—should be identified as such in the Methods section of the paper, with rationale.

2. You have noted restrictions apply regarding the availability of your data. PLOS Medicine requires that the de-identified data underlying the specific results in a published article be made available, without restrictions on access, in a public repository or as Supporting Information at the time of article publication, provided it is legal and ethical to do so. Please see the policy at 

http://journals.plos.org/plosmedicine/s/data-availability

and FAQs at 

http://journals.plos.org/plosmedicine/s/data-availability#loc-faqs-for-data-policy

3. Please clearly state somewhere in the main text the primary outcome of the study, and secondary outcomes if applicable (e.g. diagnosis of dementia or death).

4. Title: Please change “risk of” to “association with” in the title, as this is an observational cohort study and we want to avoid any implication of causality.

5. Abstract Background: The final sentence should clearly state the study question.

6. Abstract Methods and Findings: Please define the years during which the study took place.

7. Abstract Methods and Findings: Please quantify the main results comparing dementia diagnoses in those with and without depression, and the sub-analyses, with p values in addition to the 95% CIs. 

8. Abstract Methods and Findings: Please include the important dependent variables that are adjusted for in the analyses.

9. Abstract Methods and Findings: In the last sentence, please describe the main limitation(s) of the study’s methodology.

10. Methods: Please indicate the form of consent obtained, or the reason that consent was not obtained (e.g. the data were analyzed anonymously).

11. Methods: For all observational studies, in the manuscript text, please indicate: the specific hypotheses you intended to test, and when reported analyses differ from those that were planned, transparent explanations for differences that affect the reliability of the study's results. If a reported analysis was performed based on an interesting but unanticipated pattern in the data, please be clear that the analysis was data-driven.

12. Results: Please provide 95% CIs and p values associated with the analysis of differences between cohort demographic factors (data presented in Table 1).

13. Results: Please provide the p values in addition to the 95% CIs for the odds ratios of depression and risk of subsequent dementia diagnosis in the non-sibling cohort (Table 2 data).

14. Results: Please provide the p values in addition to the 95% CIs for the odds ratios of depression and risk of subsequent dementia diagnosis in the sibling cohort (Table 3 data).

15. Results: Please provide the p values in addition to the 95% CIs for the odd ratios of the associations between depression severity/type and dementia diagnosis (Table 4 and 5 data).

16. Results: For all the adjusted analyses described, please also provide the unadjusted analyses.

17. Figure 1: Please indicate in the figure caption the meaning of the black and gray lines. Please display the 95% CIs

18. Table 1: Please provide statistics to illustrate the factors for which the individuals with and without depression were significantly different, as stated in the Results.

19. Table 2 and Table 3: “Confident intervals” should be “confidence intervals” in the Table legend. Please provide p values along with the 95% CIs for all comparisons. Please define the abbreviations “OR” and “CI” used in the tables.

20. Table 4 and Table 5: For the adjusted analyses described, please also provide the unadjusted analyses. “Confident intervals” should be “confidence intervals” in the Table legend. Please define the abbreviations “OR” and “CI” used in the tables. Please provide p values along with the 95% CIs for all comparisons.

21. Table 5: The footnotes denoted by * and ** are repeated.

22. Please use the "Vancouver" style for reference formatting, and see our website for other reference guidelines:

https://journals.plos.org/plosmedicine/s/submission-guidelines#loc-references

23. Please ensure that the study is reported according to the STROBE guideline (or the most relevant guideline: http://www.equator-network.org/), and include the completed STROBE checklist as Supporting Information. When completing the checklist, please use section and paragraph numbers, rather than page numbers. Please add the following statement, or similar, to the Methods: "This study is reported as per the Strengthening the Reporting of Observational Studies in Epidemiology (STROBE) guideline (S1 Checklist)."

Comments from the reviewers:

Reviewer #1: The work uses routinely-collected data sources (EHR records from a country with UHI) to ascertain timing and incidence of dementia and depression, and then estimates the differences in risk of a record of dementia according to a previous diagnosis of depression. My role in the review process is as a statistical reviewer, and I have made some comments on the interpretation of the results with regard to the data sources as well. I can appreciate the work that has gone into this study, creating matched cohorts of this size from EHR data poses many challenges.

Statistical Issues:

The first statistical issue comes from the comparison of odds ratios between the general population cohort and the sibling-matched cohort. The incidence of the key outcome is different in the two populations (9,802/238,772 in the pop cohort, vs 1,161/50,644 in the sibling cohort). The odds ratio is not collapsible, so that a similar odds ratio between the two cohorts at the same time after diagnosis may actually reflect dissimilar relative risk (or a dissimilar odds ratio, and a similar relative risk). This issue affects comparison of ORs over time, and between single vs. multiple episodes, men vs. women etc. Estimating a rate ratio (estimated by log-binomial or log-Poisson with robust SE) is a way to avoid this issue, as is estimating relative risk from marginal estimates of risk (e.g. predictive margins in Stata).

The second statistical issue relates to the selection of covariates. What criteria was used for the selection of comorbidities subsequently used as adjustments for confounding? P-values for univariable association or effect size? Univariable pre-filtering can have unintended consequences - there is a good review by Heinz and Dunkler (Transplant International 2017; 30: 6-10) on this subject.

Data sources:

The foundation of this work is the quality of the data collection used to identify the outcome, exposure, and the potential confounders. Is there a key reference that about the timing and composition of data collection in the SNPR? It looks like it is composed of hospital and specialist care - is there general practice (family practice) collection included? Many of the PLoS Medicine readers won't be familiar with the health system behind the data collection and may need some context to understand what is accomplished in this work.

Was there missing data in the original data collection that precluded matching? How many records couldn't be used if so?

There is good evidence overall the SIR data accurately captures many conditions (from Ludvigsson et al 2011) and that previous validation studies have shown that a general dementia diagnosis is captured accurately. How well is depression exposure captured if the records are from acute (hospitalised) and specialist care episodes? In my country the majority of depression is treated in general practice (family practitioner) or by clinical psychologists and only severe cases of depression would be treated as an in-patient or by a psychiatrist. Is this a similar situation to Sweden? Are there any validation studies for PPV and sensitivity of timing of depression from the SNPR for depression of differing severity? 

There is clearly a time-dependence, and the odds of dementia diagnosis are very high immediately following a diagnosis of depression. This is hinted at in the discussion but I think needs to be discussed clearly - could the depression actually an unrecognised sign or symptom of early dementia (potentially reversing the causal mechanism) or is it due to confounding according to more intense health service use increasing the diagnosis of both diseases? 

Are there any known confounders of the depression-dementia relationship unavailable in the SNPR data? What level of confounding would be needed to account for the observed relationship? I have found the E-value is a useful tool (Ann Intern Med. 2017;167(4):268-274.) to consider how sensitive an observed outcome-exposure is to an unmeasured confounder.

Specific comments:

p2, Conclusions. Logistic regression is used throughout this manuscript, so 'increased odds' of dementia is a more accurate description of the results. 

p2, Conclusions. The phrasing part of this sentence 'of especially vascular dementia' is unclear. I would either just refer to dementia generally or reword this sentence to include the additional information about vascular dementia. 

P15, Household income. Is there a way to indicate what the relative difference in household income is for international readers (i.e. converted to USD/Euros or an indication of household income distribution in Sweden at the time of the data collection)? The difference in household income between those diagnosed with depression and not diagnosed with depression in the sibling cohort looks extreme - but this could just be my unfamiliarity with the currency.

P16, Title of table. Although it is possible to identify the source of the data from the reported N, it would be easier if the source of the data (e.g. Population matched cohort) was also included as in Table 3. 

P19, Table 5. There is a 'ns' indicator for single depressive episode and 10-20 years after diagnosis, but this doesn't appear elsewhere in the table where the 95% CI overlaps with 1. I would suggest to just remove the 'ns' as the CI already gives this information. 

Reviewer #2: PMEDICINE-D-19-03130 "Depression and the risk of subsequent dementia diagnosis: A Nationwide Cohort Study" 

In this study the authors examined the association between depression and dementia and reported depression to be associated with "an increased risk of especially vascular dementia, even

more than 20 years after the depression, and the association remained after

adjustment for familiar factors."

The data in the study come from two cohorts formed of all individuals above 50 years, living in Sweden. The first cohort was composed of individuals diagnosed with depression matched to controls without depression (n=238,772). The second was a co-sibling cohort (n=50,644) consisting of same-sex full sibling pairs with discordant depression status. This is an impressive study design, and the size of the study sample would lead us to believe that the results would be robust. Nevertheless, the following points need further reflection.

1.Novelty

The authors say

"The association between depression and subsequent dementia diagnosis has been established

in at least four independent meta-analysis indicating close to a twofold risk of developing

dementia followed by depression [7-10], as well as several reviews [11-14]."

Given the meta-analyses and reviews on this topic, the contribution of this study is unclear.

2.Depression diagnosis

Diagnosis of depression came from the Swedish National Patient Register. The manuscript provides the ICD codes for diagnosis but no detail on the register itself. Further information is required on the sources of data used in this register, in terms of consultations - family doctor, psychiatrists, hospitals? Is the diagnosis based on prescription of antidepressants or hospitalization for depression? This information is needed to understand what is being measured by the ICD codes. The target population was 3.3M and only 119,386 cases of depression were identified - this is a surprisingly low rate. 

3.Matched controls 

In the first cohort, controls were matched for age, sex, and citizenship but Table 1 shows cases and controls to differ systematically on sociodemographic and health status. It is possible that the results can be explained by systematic differences in these two groups that have not much to do with depression. 

4.Dementia diagnosis

Further details are required on how dementia was diagnosed in both cohorts. Do the data on dementia come primarily from mortality registers? Both cohorts in the manuscript have a follow-up of over 20 years but the rate of dementia is 4.1% and 2.3%. This is again surprising low. Is it possible that most cases of dementia were missed in the register?

5.Conclusion

I do not think that the conclusion "The results of the present nationwide study indicated an increased risk for dementia diagnosis after depression, even when the depression occurred 20 years or more before the dementia diagnosis" is supported by the data presented in the manuscript. Figure 1 clearly shows reverse causation (or depression being a prodromal feature of dementia) as the excess risk drops at Year 5 of follow-up and remains unchanged thereafter. There are problems with measurement of exposure and the outcome, and the selection of the control group. Furthermore, the analysis of the 20-year follow up is based on a small number of cases (481 cases in 26,044 persons in cohort 1, and 76 cases in 6,296 persons in cohort 2), making it difficult to draw conclusions.

Reviewer #3: This study on risk of dementia among older depressed adults should be of substantial interest to readers of the journal.

In the Introduction, the authors note the high prevalence of dementia and the need to identify modifiable dementia risk factors. They briefly cite literature noting the increased risk of dementia among those with depression, noting that depression may be seen as a discrete risk factor or as a dementia prodrome. 

In the final introductory paragraph, the authors note that the purpose of the study was to investigate if the association between depression and later dementia risk is time dependent, identify factors that influence the association, and whether the association is influenced by severity and frequency of depression. Given their prior literature review in the preceding paragraphs, one wonders whether the authors have any a priori hypotheses in these three areas, or if this is truly a completely exploratory study, which would require appropriate statistical correction. If the authors did have a priori hypotheses, they should state them.

In the Materials and Methods section, the authors describe reasonably well how the cohorts for the present study were derived. They refer to “depression” as opposed to “major depression.” If indeed they are focusing on major depression, they should be more precise. This is also the case for the next section, “Procedure,” in which they specify ICD-10 codes of F32 and F33 but only mention “diagnosis of depression.”

They note that “all diagnoses were set in specialist care.” It is not clear what that means. 

In the statistical methods section, the authors quickly jump to time dependency analyses without first indicating what initial analysis, if any, that they undertook comparing depression and dementia risk. They should specify how the dependent and independent variables were constructed. They should justify why separate analyses were performed for men and women, as opposed to, for example, doing separate analyses for different age cohorts, given differences in typical ages of onset for Alzheimer’s disease and vascular dementia.

It is not clear what the following sentence means: The knots were set in default positions.

If this is an exploratory study without hypotheses, then it is not clear that appropriate statistical corrections were employed.

The results, in the text, tables and figures, are clearly presented. 

In the Discussion, the authors summarize some of the main findings, but should mention their findings related to severity as well.

They then place their study in the context of prior literature. There are a few key studies that they should consider including:

Barnes et al. Arch Gen Psychiatry. 2006;63(3):273-279. Study focusing on depression as risk for MCI

Steffens et al. Biol Psychiatry. 1997;41(8):851-856. Large epidemiological study noting depression as risk for AD only when there is a few years between depression onset and dementia onset

Kokmen et al. Neurology. 1991;41(9):1393-1397. Large population study linking episodic depression and AD.

Becker et al. Am J Geriatr Psychiatry. 2009;17(8):653-663. A community based study that found that depressed mood is not a risk factor for incident dementia.

Steffens et al. Am J Geriatr Psychiatry. 2007 Oct;15(10):839-49. Clinical study noting depression as risk for non-AD dementias.

Inclusion of some of these and other studies might allow for a fuller discussion of their findings.

The authors identify appropriate limitations in their study. If this truly was an exploratory study (without a priori hypotheses), that might also be considered a limitation.

[LINK]

---

## [Decision Letter · Decision Letter 1]

25 Nov 2019

Dear Dr. Nordström,

Thank you very much for re-submitting your manuscript "Depression and the association with subsequent dementia diagnosis: A Nationwide Cohort Study" (PMEDICINE-D-19-03130R1) for review by PLOS Medicine.

I have discussed the paper with my colleagues and the academic editor and it was also seen again by three reviewers. I am pleased to say that provided the remaining editorial and production issues are dealt with we are planning to accept the paper for publication in the journal.

[LINK]

We look forward to receiving the revised manuscript by Dec 02 2019 11:59PM. 

Sincerely,

Caitlin Moyer, Ph.D.

Associate Editor 

PLOS Medicine

plosmedicine.org

Requests from Editors:

1.Response to reviewers: Thank you for your response to Reviewer 1, point 3. However, please include in the Discussion section (lines 329-344) as a limitation the fact that because general/family practice data are not included in the SNPR collection, there is a possibility that some of the individuals included in the study may have been diagnosed with dementia/depression in a primary care setting. Please also include in your discussion that the fact that the diagnoses were made in specialty care may speak to lower than expected rates of dementia/depression in the cohorts (in reference to the points brought up by Reviewer 2).

2.Data Availability Statement: Thank you for your response regarding your limitations on making the study data available. If the data are owned by a third party but freely available upon request, please note this and state the owner of the data set and contact information for data requests (web or email address). Note that a study author cannot be the contact person for the data. If the data are not freely available, please describe briefly the ethical, legal, or contractual restriction that prevents you from sharing it. Please also include an appropriate contact (web or email address) for inquiries (again, this cannot be a study author).

3.Title: Please include: “A Swedish nationwide cohort study” to reflect the setting where the study took place, and also please include the dates of the study in the title.

4.Author summary: Please use bullets for each point in the Author Summary.

5.Author summary: “Why was this study done?”: In the first point, please clarify what is meant by “...leading cause of dependency…”

6.Author summary: “What did the researchers do and find?”: Lines 87-89, please clarify to “...after the diagnosis of depression” in both sentences.

7.Author summary: “What did the researchers do and find?”: Line 91: Please delete the word “one”

8.Author summary: “What do these findings mean?”: Please revise “after a depression” to “after the diagnosis of depression”

9. Introduction: Line 110: Please revise “...has been established…” to “...has been suggested…” or similar, because it is difficult to say something was established in a meta-analysis or review.

10.Methods, and Table 1: Please clarify in the methods for the sibling cohort whether some sibling pairs were discordant for citizenship (as indicated by the mismatched counts for Swedish and non-Swedish citizenship in Table 1 and the chi-squared test.

11.Methods: Lines 219-220: Please add the ethical approval number (e.g. Nr 2013-86-31) to the sentence describing the ethical approval. 

12.Results: Line 238: Please also provide the standard deviation for the mean follow-up time.

13.Discussion: Paragraph 1: Line 270: Please revise to “...while adjusting for a set of covariates, including…” and deleting the phrase “a rich set” as this does not convey any specific information regarding the covariates. 

14.Discussion: Line 272 (and throughout discussion where the term “significant” is used): Please clarify what you mean by "significant". If statistical significance is intended, please indicate that.

15.Table 1: Please define the abbreviation “SD” in the table legend. 

16.References: Please use the "Vancouver" style for reference formatting, and see our website for other reference guidelines: https://journals.plos.org/plosmedicine/s/submission-guidelines#loc-references

17.Thank you for including the STROBE checklist. Please revise the checklist, using section and paragraph numbers, rather than page numbers, to refer to locations of checklist items.

Comments from Reviewers:

Reviewer #1: Thank you for the revised manuscript and responses you have given to mine and the other reviewers comments. 

The main areas of concern from the original manuscript were the non-collapsibility of odds ratios, selection of covariates, and the suitability of the data for the research questions.

The selection of covariates has been clarified and this no longer a concern. The limitations to the data has been more clearly explained, the one area I feel that needs to be articulated is that the measurement of depression is most likely to be limited to acute cases, and so that only severe depression may be a risk factor for subsequent development of dementia. There is a note in the methods about the recording of major depression may exclude primary care visits - it should be made clear that the exposure measured here is the record of a diagnosis of major depression by a specialist and so is likely to be a record of severe cases. The amount of data excluded for missingness is small and not of concern given the final sizes of the cohorts.

You are correct that the issue with non-collapsibility is more severe as prevalence increases. Adding the incidence to T2 and T3 is reassuring and it does appear the non-collapsibility will have a limited impact here, and the appropriate terminology (risk -> odds) is used. I personally prefer that a risk ratio be estimated as it makes for a clearer evidence, but I think the changes are sufficient that if this is added as a minor limitation to the manuscript I think it is acceptable to present ORs over RRs.

Reviewer #2: "Depression and the association with subsequent dementia diagnosis: A Nationwide Cohort Study" # PMEDICINE-D-19-03130R1

I have the same concerns with the revised manuscript as that with the previous version.

1.Novelty

There are several meta-analyses on the association between depression and dementia, the authors do not make a clear case for what is new in yet another study.

Re reverse causation, the authors ought to look at this paper, https://www.ncbi.nlm.nih.gov/pubmed/28514478

2.Depression diagnosis

The authors say that depression rates are low because diagnosis of depression is drawn from "specialist care". What are the implications for generalizability? I would also recommend that this detail be indicated in the title, abstract, and conclusions of the paper.

3.Dementia diagnosis

The low rate of dementia is explained by the authors as being due to diagnosis in "specialist care". The implications for generalizability ought to be discussed in the limitations section.

4.Conclusion

I do not think that the conclusion "…depression diagnosis is significantly associated with increased risk of dementia diagnosis, even when depression occurs more than 20 years before the dementia diagnosis" is supported by the data presented in the manuscript. Figure 1 clearly shows reverse causation (or depression being a prodromal feature of dementia) as the excess risk drops at Year 5 of follow-up and remains unchanged thereafter. 

Reviewer #3: The authors have addressed my concerns.

[LINK]

---

## [Editor Report · Decision Letter 2]

13 Dec 2019

Dear Dr. Nordström, 

On behalf of my colleagues and the academic editor, Dr. Carol Brayne, I am delighted to inform you that your manuscript entitled "Depression and the association with subsequent dementia diagnosis: A Swedish Nationwide Cohort Study from 1964 to 2016" (PMEDICINE-D-19-03130R2) has been accepted for publication in PLOS Medicine. 

PRODUCTION PROCESS

PRESS

PROFILE INFORMATION

Thank you again for submitting the manuscript to PLOS Medicine. We look forward to publishing it. 

Best wishes, 

Caitlin Moyer, Ph.D.

Associate Editor 

PLOS Medicine

plosmedicine.org